# Transcriptome analysis of *Curcuma wenyujin* from Haikou and Wenzhou, and a comparison of the main constituents and related genes of Rhizoma Curcumae

**Lilan Lu**[1,2]⊗*, **Peiwei Liu**[1]⊗, **Yanfang Yang**[3]⊗, **Yuxiu Zhang**[1], **Caixia Wang**[4], **Jian Feng**[1], **Jianhe Wei**[1]*

**1** Haikou Provincial Key Laboratory of Resources Conservation and Development of Southern Medicine, Haikou Branch of the Institute of Medicinal Plant Development, Chinese Academy of Medical Sciences and Peking Union Medical College, Haikou, China, **2** Hainan Key Laboratory of Tropical Oil Crops Biology/ Coconut Research Institute, Chinese Academy of Tropical Agricultural Sciences, Wenchang, China, **3** State Key Laboratory of Tree Genetics and Breeding, Key Laboratory of Tree Breeding and Cultivation of State Forestry Administration, The Research Institute of Forestry, Chinese Academy of Forestry, Beijing, China, **4** Institute of Agricultural Environment and Soil, Hainan Academy of Agricultural Sciences, Haikou, China

⊗ These authors contributed equally to this work.
* lulilan1234@163.com (LL); wjianh@263.net (JW)

**Data Availability Statement:** All relevant data are within the manuscript and its Supporting information files.

## Abstract

For more than a thousand years, Rhizoma Curcumae (known as *E zhu*), a Chinese herbal medicine, has been used to eradicate blood stasis and relieve aches. The plant *Curcuma wenyujin*, which is grown primarily in Wenzhou, China, is considered the best source of Rhizoma Curcumae. In this study, we sought to ascertain differences in transcript profiles of *C. wenyujin* grown in traditional (Wenzhou) and recently established (Haikou) production areas based on Illumina and RNA (RNA-seq) sequencing. We also examined differences in the main components of the volatile oil terpene; curcumin, polysaccharide, and starch constituents and related genes in the corresponding pathways, in *C. wenyujin* cultivated in the two production areas. We accordingly found that the essential oil (2.05%), curcumin (1.46%), and polysaccharide (8.90%) content in Wenzhou rhizomes was higher than that in the rhizomes of plants from Haikou (1.60%, 0.91%, and 6.15%, respectively). In contrast, the starch content of Wenzhou rhizomes (17.0%) was lower than that of Haikou rhizomes (23.8%). Furthermore, we detected significant differences in the oil components of Haikou and Wenzhou rhizomes, with curzerene (32.34%), curdione (21.35%), and germacrene B (9.39%) being the primary components of the essential oil derived from Wenzhou rhizomes, and curzerene (20.13%), curdione (14.73%), and cineole (9.76%) being the main constituents in Haikou rhizomes. Transcriptome and qPCR analyses revealed considerable differences in gene expression between Wenzhou and Haikou rhizomes. The expression of terpene, curcumin, and polysaccharide pathway-related genes in Wenzhou rhizomes was significantly up-regulated, whereas the expression of starch-associated genes was significantly down-regulated, compared with those in Haikou rhizomes. Difference in the content of terpene, curcumin, polysaccharides, and starch in rhizomes from the

**Funding:** YES, Hainan province Key Scientific and Technological Research and Development Special Project (Grant No: ZDKJ2016006), The funders had a role in study design, data collection and analysis, decision to publish, or preparation of the manuscript.

**Competing interests:** The authors have declared that no competing interests exist.

two production areas could be explained in terms of differences in expression of the related genes.

## Introduction

*Curcuma* is a plant genus comprising more than 70 rhizome-producing species worldwide. In China, there are approximately 20 *Curcuma* species, a small number of which have been used in Chinese herbal medicines and food products for more than a thousand years. Rhizoma Curcuma is the dried rhizome of *Curcuma kwangsiensis* S.G. Lee & C.F. Liang, *Curcuma phaeocaulis* Valeton, and *Curcuma wenyujin* Y.H. Chen & C. Ling [1]. The essential oils of *Curcuma* are considered the active ingredients of these plants and have been demonstrated to have antiviral and antitumor properties [2–5]. The main constituents of the essential oils in Curcumae rhizomes are curcumol, germacrone, and curdione, which account for most of the known pharmacological effects, and are typically used as quality control markers for the essential oils [6–8]. *C. wenyujin* ("*wen e zhu*" in Chinese) from Wenzhou, Zhejiang Province, China, is widely considered to be the best source of Rhizoma Curcumae ("*e zhu*" [9–11]), on account of its distinctive regional characteristics and high medicinal value, and is mainly produced in Ruian, Wenzhou. The main active constituents of *C. wenyujin* are volatile oil and curcumin [12, 13], the former of which is composed of terpenoids and sesquiterpene derivatives, which have been reported to have anti-inflammatory, antitumor, lipid-regulating, antioxidant, antiviral, antibacterial, and other therapeutic effects [14–17].

*C. wenyujin* is mainly cultivated in China, wherein the quality of the rhizomes varies substantially depending on the variety, cultivation region, cultivation techniques, and extraction methods [11, 18, 19]. *C. wenyujin* cultivated in different regions, for example, the traditional and non-traditional producing areas, can differ notably with respect to oil components. Given that different environmental conditions can promote changes in the active constituents and alter the transcriptional regulation of genes that control the formation of secondary metabolites, the same *C. wenyujin* genotype can produce rhizomes with differing qualities depending on the region in and conditions under which the plants are cultivated, and consequently this can potentially present difficulties with respect to quality control.

A notable characteristic of plants is their plasticity in response to unfavorable environmental conditions [20, 21]. Plants typically initiate a series of specific biochemical responses to environmental stressors associated with diverse aspects of their biology, including development, evolution, molecular biology, anatomy, physiology, genetics, and biochemistry [22, 23], which are of particular importance from the perspectives of predicting changes in community composition, distribution, and crop productivity in response to continuously changing environments [24, 25].

A range of environmental factors (both biotic and abiotic) can affect (often adversely) medicinal plants during their growth, thereby influencing the formation and accumulation of terpenoids, ketones, phenols, alkaloids, flavonoid, flavonols, volatile oils, and other secondary metabolites [26, 27]. For example, differences in the chemical and physical properties of soils (determined by the parent material) can influence the growth and quality of plants, and in this regard, it has been found that the oil content of *C. wenyujin* rhizomes cultivated in alkaline soils is higher than that of rhizomes cultivated in acidic red soils [28].

Isopentenyl pyrophosphate (IPP) is an important intermediate in the biosynthesis of terpenoids and steroids in plants, and steroidal alkaloids can also be formed through other

pathways. IPP is formed as a common intermediate in both the mevalonate (MVA) and 2-methyl-D-erythritol-4-phosphoric acid (MEP) pathways, and can pass through cell walls to transfer between these two pathways [29]. Abiotic environmental factors (including light, temperature, moisture, and nutrition) affect metabolism and the synthesis of metabolites, predominantly by influencing gene expression and transcriptional factors in the associated metabolic pathways (e.g., MVA and MEP) of secondary metabolites, such as terpenoids, flavonoids and flavonols [30–38]. Thus, it is speculated that variations in natural environmental conditions in the Wenzhou and Haikou regions, which are characterized by subtropical monsoon and tropical monsoon climates, respectively, may contribute to notable differences in the secondary metabolite profiles of *C. wenyujin* rhizomes sourced from these two regions.

In recent years, drug companies (such as Haikou Bikai Pharmaceutical Co. Ltd) in Haikou, Hainan Province, have introduced the local cultivation of *C. wenyujin* using plant material originally sourced from Ruian, Wenzhou, Zhejiang Province [39]. *C. wenyujin* is cultivated in Haikou for the production of medicines derived from this plant, with the goal of meeting the growing demand for *C. wenyujin* constituents (volatile oils, crude extracts, and raw materials). In this study, we sought to determine the main constituents and genetic characteristics of *C. wenyujin* rhizomes sourced from Wenzhou and Haikou, with the aim of assessing the qualities of the respective *C. wenyujin* rhizomes. The main objective was to analyze the content of essential oils (such as terpene), curcumin, polysaccharides, and starch in *C. wenyujin* rhizomes obtained from the traditional production areas (Wenzhou) and newly established production areas (Haikou). In this regard, although there have been a few transcriptome studies on plants in the ginger family (e.g., *Curcuma zedoaria* (Christm.) Roscoe) [40–42], to date there have been no studies that have examined genetic variations in the metabolic pathways of *C. wenyujin*. Moreover, there have been only a limited number of studies on the transcriptome of *C. wenyujin* and the mechanisms underlying interspecific differences in the primary and secondary metabolites that accumulate in the rhizomes of *C. wenyujin* rhizomes cultivated in traditional and non-traditional production areas.

In this study, using Illumina and RNA (RNA-seq) sequencing, we examined the transcript profiles of *C. wenyujin* rhizomes produced in the traditional production area (Wenzhou) and those derived from an area in which the plant has recently been introduced (Haikou). We also sought to determine variations in the content of the main components of the volatile oil terpene, curcumin, polysaccharides, and starch, and related genes in the corresponding pathways. The data, thus, obtained provide a basis for evaluating the quality of *C. wenyujin* and elucidating the molecular mechanisms underlying the biosynthetic pathways in plants cultivated in these two production areas, and hence will enable us to assess the potential medicinal value, development, and utilization of *C. wenyujin* in more agriculturally productive areas, such as Haikou.

## Materials and methods

### Plant materials

Samples of the traditionally cultivated (Wenzhou) *C. wenyujin* (WZ) were purchased from Taoshan Town, Ruian, Wenzhou, Zhejiang Province, on December 5, 2017, whereas those of the introduced (Haikou) *C. wenyujin* (HK) were obtained on the same date from cultivation fields at the Haikou branch of the Institute of Medicinal Plants. The cultivation management of *C. wenyujin* in the two areas is similar, and details of the environmental and weather conditions of these areas are presented in S1 Table. The *Curcuma* samples were subsequently identified as *C. wenyujin* Y.H. Chen & C. Ling. by Li-Rongtao, Associate Professor at the Institute of

Medicinal Plants, Chinese Academy of Medical Sciences. For the purposes of the present study, we selected the rhizomes of the *C. wenyujin* plants as the experimental materials.

## RNA isolation, quantification, and qualification

Total RNA was isolated from each sample (rhizomes) using a Quick RNA isolation kit (Bioteke Corporation, Beijing, China). Three biological replicates per experimental group were used for sequencing. Agarose gel (1%) electrophoresis was used to assess the contamination and degradation of RNA. The purity and concentration of the isolated RNA were determined using a Nano spectrophotometer (Photometer®; IMPLEN, CA, USA) and a Qubit®2.0 Fluorometer Kit (RNA Assay, Life Technologies, CA, USA), respectively, whereas an Agilent 2100 Bioanalyzer system (Nano 6000; Agilent Technologies, CA, USA) was used to evaluate RNA integrity.

## Library preparation and sequencing

An RNA library was prepared using a total of 3 μg RNA per sample, which was analyzed using high-throughput Illumina sequencing technology at the Beijing BioMarker Technologies Company (Beijing, China). An RNA Library Prep Kit (NEB for Illumina, USA) was used for generating sequencing libraries, in accordance with the manufacturer's recommendations, and the sequences of each sample were attributed using index codes. mRNA was extracted from the total RNA using Poly-T oligo-attached magnetic beads. Fragmentation of the sequences was facilitated by crushing the samples in the presence of divalent cations in a NEB-Next first-chain synthetic buffer (5×) at an elevated temperature. First-chain cDNA synthesis was performed using RNase-H (Invitrogen, USA), and Polymerase I (Invitrogen, USA) and RNase-H (Invitrogen, USA) were used for second-chain cDNA synthesis. The remaining overhangs were blunt-ended based on polymerase or exonuclease reactions. Subsequent to adenylation of the DNA fragments (3´ ends), the junction (NEBNext), with a hairpin loop structure, was ligated in preparation for hybridization. To preferentially select cDNA fragments of 150–200 bp in length, we used an XP system (AMP, Beckman Coulter, Beverly, USA) to separate the DNA fragments. Prior to PCR, 3 μL USER enzyme (NEB, USA) was added to the samples containing the size-selected adaptor-ligated cDNA, which were heated at 37˚C for 15 min, followed by 95˚C for 5 min. PCR was carried out using DNA polymerase (Phusion High-Fidelity), Index (X), and Universal Primers (PCR), and the XP system (AMP, Beckman Coulter, Beverly, USA) was applied to purify the PCR products. Library quality was evaluated using an Agilent 2100 Bioanalyzer system (Nano 6000; Agilent Technologies, CA, USA).

## Data analysis and assembly

An internal Perl script was initially used to handle the raw read data in fastq (fq) format. Clean reads were acquired by deleting reads including adapters, poly-N sequences, and low-quality reads from the raw data. Moreover, we statistically assessed the sequence repetition, GC content, and Q20 and Q30 levels of clean read data. For all library samples, the left read1 and right read2 files were placed in two larger files (left fq and right fq, respectively). Assembly of the transcriptome was performed based on left/right fq data using the Trinity method, with min_kmer_cov set to 2 and all other parameters automatically set to default [43, 44].

## Sequence annotation and functional classification

Genes with unique sequences (unigenes) were annotated and functionally classified based on Clusters of Orthologous Groups of proteins (COG), KOG, and eggNOG protein sequence data

libraries for manual annotation and reviewed against Swiss-Prot. Gene Ontology (GO), and Kyoto Encyclopedia of Genes and Genomes (KEGG) databases [45].

**Quantification of gene expression.** RSEM was applied to analyze the levels of gene expression in each sample [46]. Clean read data were mapped onto the assembled transcriptome, and the read counts per gene were obtained from the transcriptome database. For the analysis of differential gene expression, we used the DESeq package of R (1.10.1). *P*-value results were assessed using the Benjamin and Hochberg method (1995) to control for the false discovery rate (FDR), and genes with a *P*-value < 0.05 were defined as being differentially expressed (DEGs) [45, 47].

**GO enrichment analysis.** For analysis of DEG enrichment, we used the R package topGO, based on the Kolmogorov–Smirnov test [45, 48].

**KEGG pathway enrichment analysis.** KEGG pathway enrichment with DEGs was determined using KOBAS 2.0 software [47, 49].

## Sample preparation and Gas Chromatography-Mass Spectrometry (GC-MS) analysis

Sample materials were obtained from large rhizomes of *C. wenyujin*, which had been boiled for 1 h at 100°C and 0.1 MPa, and then cut into 1–2 cm slices and dried for 24 h at 50°C. The volatile oil content of samples was then extracted in accordance with Chinese Pharmacopoeia guidelines [1]. For high-performance liquid chromatography (HPLC) we used an Agilent 7890A gas chromatograph under the following chromatographic conditions: DB-5MS column (0.25 mm × 30 m, 0.25 μm), helium carrier gas, EI ion ionization method, 200°C source temperature, 70 eV electron energy, 250°C interface temperature, 150 μA emission current, 35 to 455 mass range, and a 0.4 s scan cycle. Xcalibur 1.2 was used as a data processing system, and the map library was NIST (Version 1.7). The heating program used was as follows: an initial temperature of 60°C held for 1 min, followed by an increase from 60°C to 100°C at 4°C·min$^{-1}$, from 100°C to 120°C at 2°C·min$^{-1}$, from 120°C to 180°C at 1°C·min$^{-1}$, and from 180°C to 230°C at 23°C·min$^{-1}$, where the temperature was held for 10 min. Other parameters were as follows: 53 kPa column pressure, 1.04°C·min$^{-1}$ flow rate, and 0.2 μL sample volume. A 20 μL volume of eugenol:*n*-hexane (20:1) was added to a 2 mL brown gas-phase flask, to which was added 1000 μL of *n*-hexane. The solution was mixed by vortexing for 1 min and then filtered. To obtain the total ion chromatograms of different volatile oils, we used an Agilent 7890A gas chromatograph coupled to a 5975 C quadrupole mass spectrometer. The relative value (percentage) of each chromatogram peak was calculated using the peak area normalization method, and following a computer search and manual analysis, the compounds were identified with reference to the literature.

## Quantitative reverse-transcription PCR (qRT-PCR) analysis

qPCR analysis was performed using an IQ5 Multicolor (CFX96) PCR detection system (Bio-Rad, Bole, USA) and a Power PCR Master Mix (SYBR green) (Takara, Beijing, China). RNase-free DNase (TaKaRa, Beijing, China) was used to extract the total RNA from the *C. wenyujin* rhizomes, and 15 primers (Oligod-T) and a SuperScript (III) RT kit (Invitrogen, USA) were used for the reverse transcription (RT) reaction. For each RT reaction, we used 20 μL reaction mixtures containing 10 μL of master mix reagent (2× SYBR Green Master) (TaKaRa, Beijing, China), 10 ng of sample (cDNA), and 200 nM of gene-specific primers. The cycle conditions were as follows: an initial denaturation at 95°C for 30 s, followed by 40 cycles at 95°C for 5 s and 60°C for 34 s. The amplification program was then followed by a 55°C–85°C melting curve analysis, with each temperature maintained for 5 s. For PCR amplification, we used

gene-specific primers (S2 Table), the annealing efficiency of which was assessed using the primer program (PRIMER 3.0) [50]. As an internal control gene, we used c121677 against which the average amplification of three replicate samples was normalized. The relative expression values of the target genes were calculated by comparing the target gene cycle thresholds (CTs) with those of the c121677 housekeeping gene using the $2^{-\Delta\Delta CT}$ method [51, 52].

## Determination of curcumin content

For HPLC analysis of curcumin, we used an Agilent Extend C18 column (4.6 × 250 mm, 5 μm) under the following chromatographic conditions: a mobile phase of 4% acetonitrile and glacial acetic acid solution (45:55), a flow rate of 1.0 mL·min$^{-1}$, a detection wavelength of 420 nm, a column temperature of 25°C, and an injection volume of 10 μL. The number of theoretical plates was not less than 4,000. The curcumin reference standard was accurately weighed (6.35 mg) after ensuring that it had been dried to a constant weight [CAS: 110823–201706; China National Institute of Food and Drug Control (CNIFDC), China], and was then placed into a 25 mL brown volumetric flask, dissolved with methanol to 25 mL, shaken, and filtered as the reference stock solution. A curcumin control solution (25.4 μg·mL$^{-1}$) was prepared by diluting a 1 mL aliquot of this stock solution with methanol in a 10 mL brown volumetric flask. Dried powdered rhizome (0.5 g) was placed into a 50 mL conical flask to which 20 mL methanol was added. The mixture was sonicated for 1 h, and then centrifuged. To the resulting supernatant, 15 mL of methanol was added, followed by sonication for 40 min and subsequent filtering. The residual material was rinsed with a small amount of methanol and refiltered. The filtrate, thus, obtained was added to the initial extraction supernatant solution, and the combined filtrate was then made up to 20 mL using methanol at a temperature below 20°C.

## Determination of polysaccharide content

The polysaccharide content of rhizomes was determined in accordance with the methodology of the "Chinese Pharmacopoeia" (Appendix VA; 2015) [1]. The extracted polysaccharides were analyzed using a UV1601 UV-Vis spectrophotometer (Shimadzu, Japan) at a wavelength of 490 nm, with an appropriate amount of anhydrous glucose being used as the reference material (reference D-anhydroglucose; CAS: 0833–9501; CNIFDC, China). The analytical chemicals, apparatus, conditions, and preparation of standard and sample solutions have previously been described in detail by Chen [53].

## Determination of starch content

Starch content was analyzed using a UV1601 UV-Vis spectrophotometer (Shimadzu, Japan) at a wavelength of 600 nm, with a soluble starch standard (chromatographic grade, CAS: 140602; CNIFDC, China). The analytical chemicals, apparatus, analytical conditions, standard preparation, starch extraction, and preparation of *C. wenyujin* rhizomes samples have previously been described in detail by Zhang et al. [54].

## Statistical analysis

Average values of each parameter were calculated from three replicate analyses, and standard errors of the mean values were obtained. Univariate analysis (analysis of variance) of variance was applied to determine the significance of the results among different treatments, with a *P*-value < 0.05 being considered significant. SPSS V.13 was used for statistical analyses.

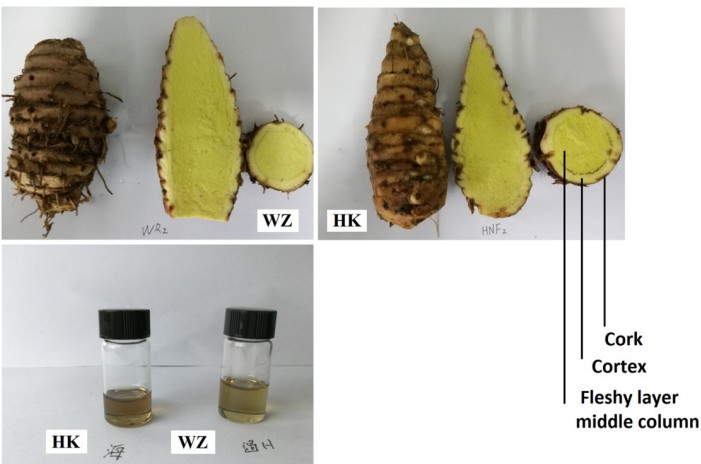

**Fig 1. The rhizomes and volatile oils of *Curcuma wenyujin* from Wenzhou and Haikou.**

## Results

### Analysis of rhizome constituent and volatile oil compositions

Our analyses revealed notable differences in the rhizome (*E zhu*) structure and volatile oil color and content of the *Curcuma wenyujin* samples obtained from the two different regions. The volatile oil content of *C. wenyujin* rhizomes derived from Wenzhou was found to be higher than that in the rhizomes from Haikou, and these oils were characterized by bright and dark yellow colors, respectively (Fig 1). Paraffin sections of rhizome tissue, stained using the periodic acid-Schiff (PAS) reaction and potassium iodide, revealed the cellular localization of oil and the quantity of starch granules, respectively (Fig 2). The oil (2.05%), curcumin (1.46%), and polysaccharide (8.90%) content of *C. wenyujin* rhizomes from Wenzhou was significantly

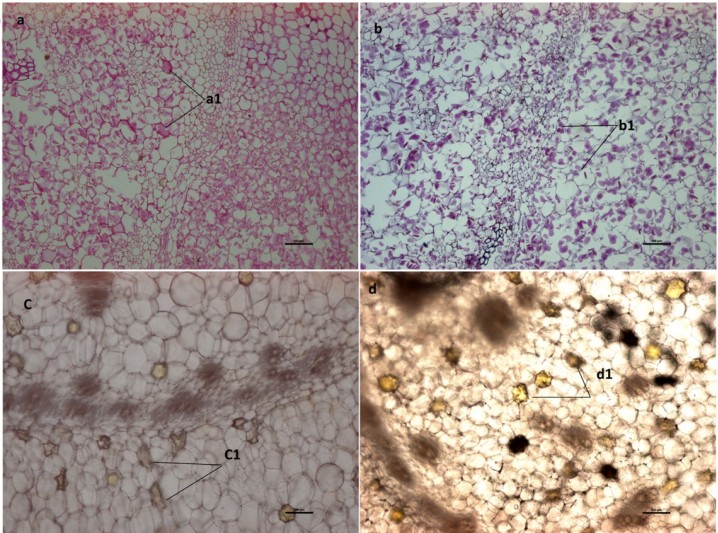

**Fig 2. Paraffin sections of *Curcuma wenyujin* rhizomes showing starch (potassium iodide staining) (a, b) and oil cells (PAS reaction) (c, d) in rhizomes from Wenzhou (b, d) and Haikou (a, c) observed under a biological microscope (a, b scale = 50 μm; c, d scale = 100 μm).**

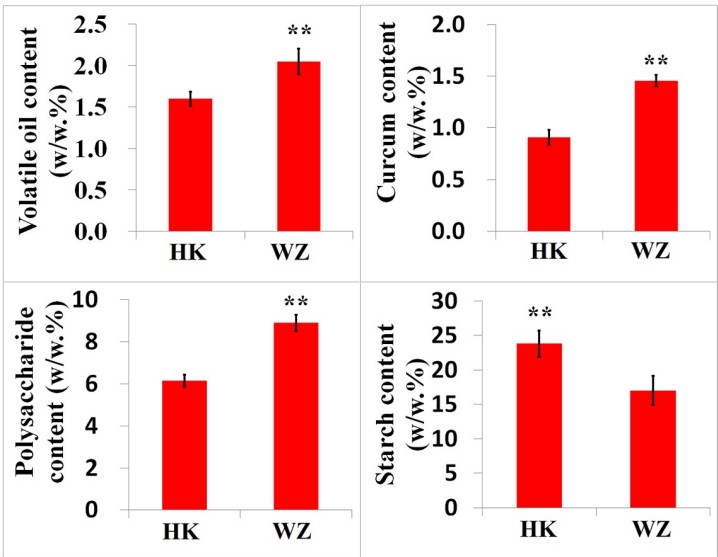

**Fig 3. The content of oil, curcumin, polysaccharide, and starch in *Curcuma wenyujin* rhizomes produced in Wenzhou and Haikou.**

higher than that of rhizomes from Haikou (1.60%, 0.91%, and 6.15%, respectively), whereas the starch content of rhizomes from Wenzhou (17.0%) was significantly lower than that of rhizomes from Haikou (23.8%) (Fig 3). We also detected significant differences in the compositions of the volatile oils derived from the rhizomes of plants obtained from the two cultivation regions, with the content of curzerene and curdione being higher in those from Wenzhou (S3 Table). Curzerene (32.34%), curdione (21.35%), and germacrene B (9.39%) were found to be the major components of the essential oil of rhizomes from Wenzhou, whereas in the rhizomes harvested from Haikou, curzerene (20.13%), curdione (14.73%), and cineole (9.76%) were identified as the main essential oil constituents. Thus, the chemical constituents of essential oils in the rhizomes of Wenzhou- and Haikou-cultivated *C. wenyujin* appear to be quite different.

## mRNA sequencing, assembly, functional annotation, and classification

To analyze the dynamics of mRNA expression in *C. wenyujin* from different production areas, we constructed and sequenced mRNA-seq libraries for *C. wenyujin* derived from Haikou (introduced cultivation) (HK) and Wenzhou (traditional cultivation) (WZ). A comparison of three replicates of the two experimental groups verified that they do not differ significantly at the sequence level (S4 and S6 Tables). Sequencing of a total of six transcriptome samples (HK1, HK2, HK3, WZ1, WZ2, and WZ3) yielded a total of 41.74 Gbp of clean data (reaching up to at least 6.37 Gbp for each sample), with Q30 percentages (number of samples with Phred quality score higher than 30) of more than 89.15% in the mRNA-seq libraries of the six samples (S4 Table). In total, we acquired 185,006 transcripts, representing 99,942 unigenes, with N50 values of 1,796 and 1,406 bp for transcripts and unigenes, respectively, and 28,375 unigenes being larger than 1 kb (S5 and S6 Tables). Searches for all unique sequences in public databases using BLASTX (E values $< 1.00^{-5}$) [55] and subsequent functional annotation implemented based on the NR, Swiss-Prot, COG, GO, and KEGG database, enabled us to confirm the identity of 51,609 unigenes (51.6%), whereas we were unable to similarly identify the remaining 48,333 unique sequences (48.4%) (S7 Table, S1 Fig).

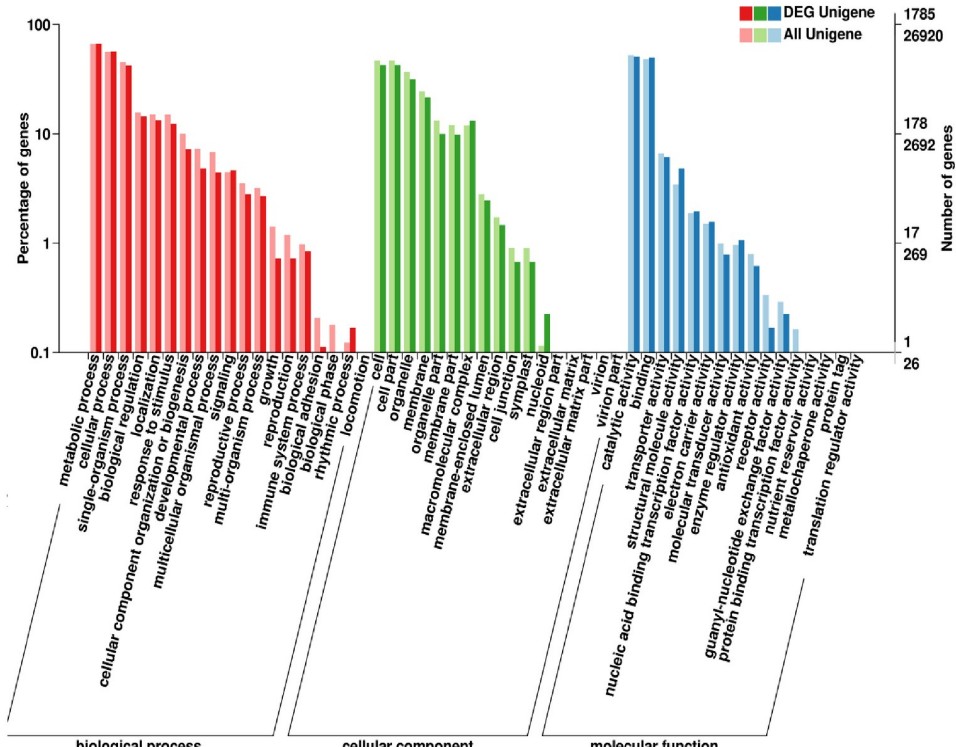

**Fig 4. A Gene Ontology (GO) histogram of the classification of annotated unigenes that were differentially expressed between Wenzhou (WZ) and Haikou (HK) rhizomes.** The red, green, and blue histograms represent the differentially expressed genes (DEGs) in the different sub-categories, whereas the light red, green, and blue histograms represent the annotated unigenes. The right-hand Y-axis represents the number of annotated genes or DEGs in the main categories, and the left-hand Y-axis represents the percentage of annotated unigenes or DEGs in the main categories.

The functions of the unique sequences extracted from *C. wenyujin* were classified based on the three GO terms cellular component (CC), biological process (BP), and molecular function (MF). Compared with HK, 15,151 unigenes and 924 DEGs in WZ were classified as CC; 19,906 unigenes and 1,276 DEGs were classified as BP; and 21,535 unigenes and 1,456 DEGs were classified as MF. The DEGs were mainly assigned to the categories "cell, go: 0005623," "cell part, go: 0044464," "organelle, go: 0043226," "membrane, go: 0016020," "cellular process, go: 0009987," "metabolic process, go: 0008152," "single-organism process, go: 0044699," "binding, go: 0005488," and "catalytic activity, go: 0003824" (Fig 4). The majority of the GO terms identified in the present study are in accordance with the GO categories defined previously [52, 56].

With respect to KEGG pathways, we identified 1,597 unique sequences (Fig 5). The predominantly enriched pathways appeared to include the following: endocytosis (36; 3.5%) and phagosome (35; 3.4%) in cellular processes; endoplasmic reticulum protein processing (70; 7.09%), spliceosome (77; 8.0%), ribosome (89; 9.09%), and RNA transport (57; 5.5%) in genetic information processing; and carbon metabolism (72; 7.12%), and biosynthesis of amino acids (58; 5.6%) in metabolism.

## Differential gene expression

We used the fragments per kilobase per million reads mapped (FPKM) procedure to calculate the expression level of genes [57], and DESeq (R package, version 1.10.1) was used for DEG

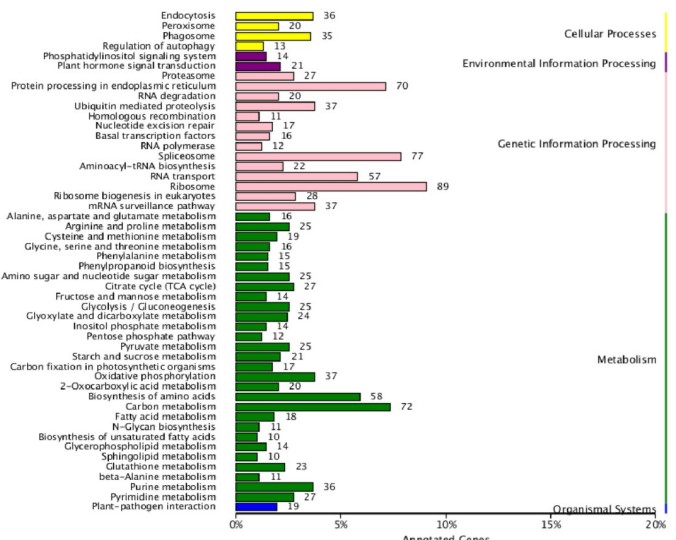

**Fig 5. Annotation of unique sequences (1,597) based on KEGG classification.**

analysis, The latter revealed that 4,620 genes were significantly differentially expressed in WZ compared with those in HK, of which 3,978 were up-regulated and 642 were down-regulated (Fig 6).

To determine similarities and differences in the transcriptional changes of genes in WZ and HK *C. wenyujin*, we performed hierarchical clustering, which accordingly revealed differences in the expression patterns of genes in the two sources of *C. wenyujin* (Fig 7). A subset of transcripts, together with their respective fold changes and false discovery rate-corrected *P*-values, for both clusters, are presented in S8 Table. Among these, genes related to terpene, curcumin, and polysaccharide pathways were found to be up-regulated in *C. wenyujin* from Wenzhou, where genes associated with starch and sugar pathways were up-regulated in *C. wenyujin* from Haikou.

## Functional enrichment analysis of DEGs

In the present study, we used topGO to analyze the functional enrichment of DEGs with respect to the three GO functional categories, CC, BP, and MF (enrichment significance, $K_S <$ 0.05), the details of which are shown in S9 Table. Among all DEGs, plastid and chloroplast (CC); methylerythritol 4-phosphate (MEP) pathway, isopentenyl diphosphate biosynthetic process, and thylakoid membrane organization (BP); and protein serine/threonine kinase activity and protein kinase activity (MF) were enriched in WZ rhizomes compared with that in HK rhizomes. Among the up-regulated DEGs, the GO terms pollen tube tip (CC); leaf morphogenesis (BP); and polyamine oxidase activity, isomerase activity, ion channel activity, and phospholipase activity (MF) were significantly enriched in WZ compared with those in HK, whereas for down-regulated DEGs, cell wall (CC); defense responses to fungus (BP); and peroxidase activity, copper ion binding, sucrose synthase activity, heat shock protein binding, and catalytic activity (MF) were significantly enriched in WZ compared with those in HK (S1 File).

Fig 8 shows the results of our COG functional classifications of the consensus sequences of genes that were differentially expressed between WZ and HK. The 1,698 genes that were differentially expressed between the rhizomes obtained from these two sources can primarily be divided into the following eight categories: general function prediction only (R) (481, 28.33%);

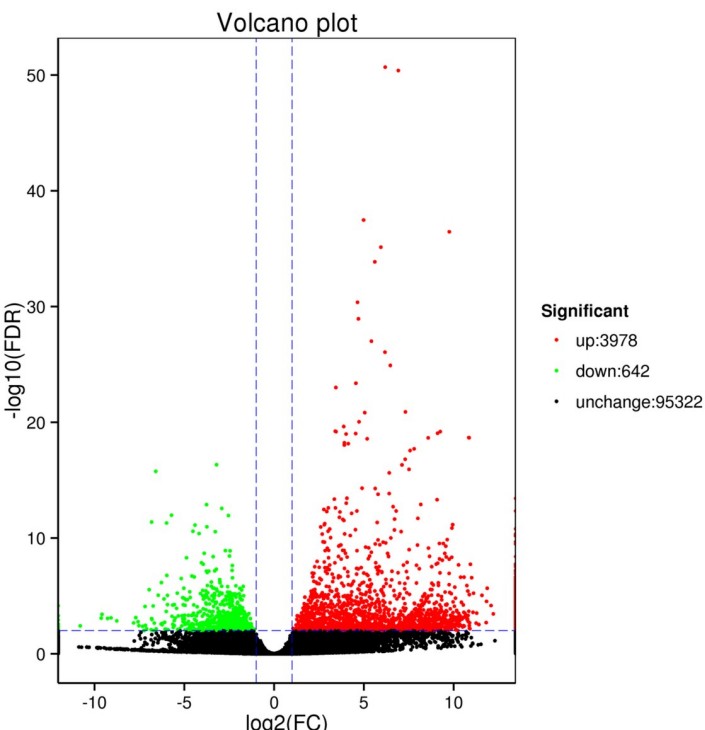

**Fig 6. Differential expression of the unique genes in *Curcuma wenyujin* rhizomes obtained from Wenzhou (WZ) and Haikou (HK).** Green and red dots denote genes with significantly different expression (FDR < 0.01), with green indicating down-regulated genes, red indicating up-regulated genes, and black indicating those genes showing no significantly different expression.

transcription (K) (230, 13.55%); replication, recombination, and repair (L) (208, 12.25%); signal transduction mechanisms (T) (190, 11.19%); translation ribosomal structure and biogenesis (J) (189, 11.13%); posttranslational modification protein turnover chaperones (O) (178, 10.48%); amino acid transport and metabolism (E) (153, 9.01%); and carbohydrate transport and metabolism (G) (140, 8.24%). The corresponding numbers and percentages of the annotated genes in these eight categories are as follows: R (2,427, 19.82%), K (415, 55.42%), L (512, 40.63%), T (405, 46.91%), J (1,426, 13.25%), O (1,373, 12.96%), E (892, 17.15%), and G (765, 18.30%). Thus, the genes associated with these eight categories can be assumed play vital roles in the growth of *C. wenyujin* in the Wenzhou traditional production area.

Similarly, functional analyses of the DEGs based on the KEGG classification of enriched pathways revealed that with respect to all genes differentially expressed between WZ and HK, 179 pathways showed enrichment, among which there were 117 and 62 pathways associated with up- and down-regulated DEGs, respectively (S10 Table). For these pathways, we analyzed the significance of the enrichment factor and Q values, and in Fig 9, we present the first 20 minimal Q value pathways. In total, 980 DEGs were assigned to KEGG pathways, among which, 844 and 136 were up- and down-regulated, respectively. The dominant and significantly enriched pathways are listed in S10 Table. With respect to all DEGs, the predominantly enriched KEGG pathways associated with the following functional areas: ribosome, spliceosome, endoplasmic reticulum protein processing, ubiquitin-mediated proteolysis, RNA transport, phagosome, glyoxylate and dicarboxylate metabolism, and glutathione metabolism. Pathways predominantly enriched with the up-regulated DEGs included spliceosome, carbon metabolism, ubiquitin-mediated proteolysis, proteasome, citrate cycle (TCA cycle),

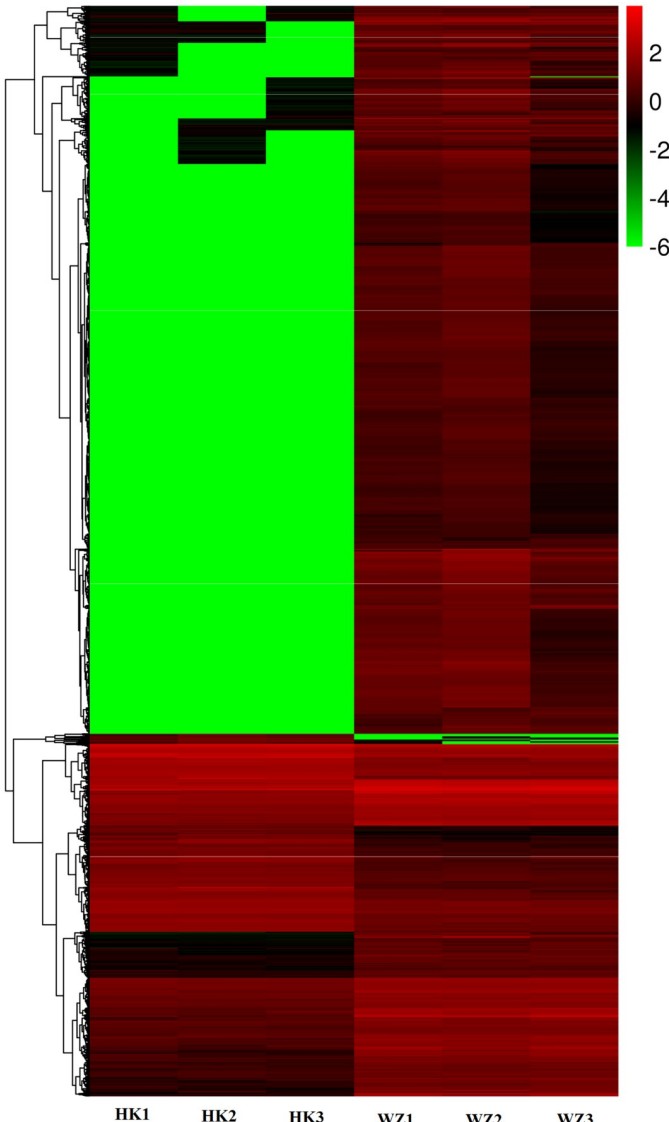

**Fig 7. Cluster analysis of genes that were differentially expressed (4,620) between Wenzhou (WZ) and HK (HK) rhizomes.** Red and green represent up- and down-regulated genes, respectively, in Wenzhou (WZ) rhizomes compared with those of Haikou (HK). The heatmap was produced based on fragments per kilobase per million (FPKM) data.

aminoacyl-tRNA biosynthesis, arginine and proline metabolism, and glutathione metabolism, whereas those significantly enriched the down-regulated DEGs included ribosome, protein export, riboflavin metabolism, endoplasmic reticulum protein processing, plant hormone signal transduction, starch and sucrose metabolism.

## Verification of DEGs

To establish the reliability of the FPKM procedure, we selected the sequences of 15 of the DEGs that are involved in terpene, curcumin, polysaccharide, and starch pathways for qRT-PCR analysis. Our findings that the overall correlation coefficient of the linear regression analysis between FPKM and qRT-PCR was 0.959 (R = 0.959) indicated a good correlation

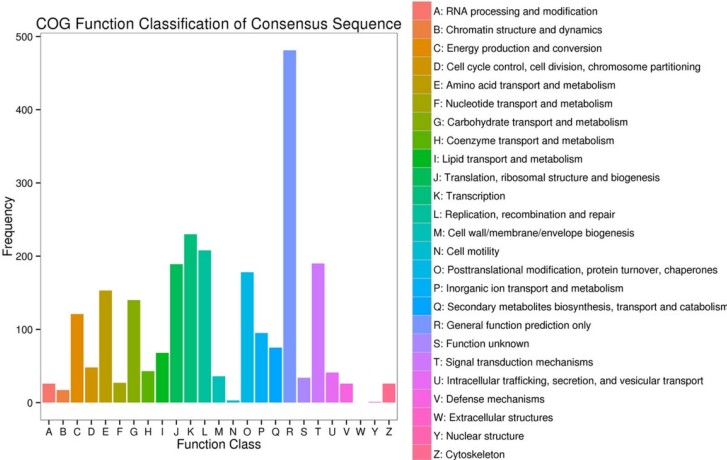

**Fig 8. Clusters of Orthologous Group (COG) functional classification of the consensus sequences of genes that are differentially expressed between Wenzhou (WZ) and Haikou (HK) rhizomes.**

between the transcript abundance detected by RT-PCR and the transcription spectrum of RNA sequence data analysis (S2 Fig). Data obtained from RNA-seq and qRT-PCR analyses revealed that those genes associated with terpene (LPS, CDS/LIS, ISPS, and TPS06), curcumin (CHS2, CHS1, and TT4), and polysaccharide (TSTA, GALE, and GH) pathways were significantly up-regulated in the rhizomes of *C. wenyujin* cultivated in Wenzhou, whereas genes associated with polysaccharide (LOC103985749) and starch (GAE, PME, BGLU, and SUS) pathways were significantly up-regulated in the rhizomes of *C. wenyujin* obtained from Haikou. These differences in gene expression patterns were consistent with those obtained based on transcriptome FPKM analysis (Fig 10).

## Transcription factor analysis

Transcription factor analysis revealed that 57 classes of transcription factors showed dynamic changes in Haikou rhizomes compared with those from Wenzhou. Compared with the former, 52 types of transcription factor were up-regulated in Wenzhou-derived rhizomes, of which the expression levels of genes in the *RLK*, *C2H2*, *Zn-clus*, *TKL*, *bZIP*, *CAMK*, *AP2/ERF*, *AGC*, *STE*, *C2C2*, *CMGC*, *C3H*, *SNF2*, *WRKY*, *bHLH*, *GNAT*, *MYB*, *HSF*, and *NAC* families were significantly up-regulated. Among these, the gene expression of certain types of transcription factor only showed up-regulation, whereas a few genes in the *RLK*, *bZIP*, *AP2/ERF*, *WRKY*, *bHLH*, *MYB*, *HSF*, and *NAC* classes also showed down-regulation in *C. wenyujin* rhizomes from WZ compared with those from HK (S3 Fig, S11 Table).

## Discussion

Environmental factors play a prominent role in determining the growth of plants, and environmental quality and climatic conditions in different growing areas are the most important factors contributing to variations in plant primary and secondary metabolites [18, 19].

Previous studies that have conducted chemical component and transcriptome analyses of *Curcuma* species have reported large variations in the content and composition of volatile oils and the content of curcumin and polysaccharide among different *Curcuma* species grown in different areas, as well as in the corresponding Rhizoma Curcumae [58–61]. Consistently, the findings of the present revealed large differences in the content and composition of volatile

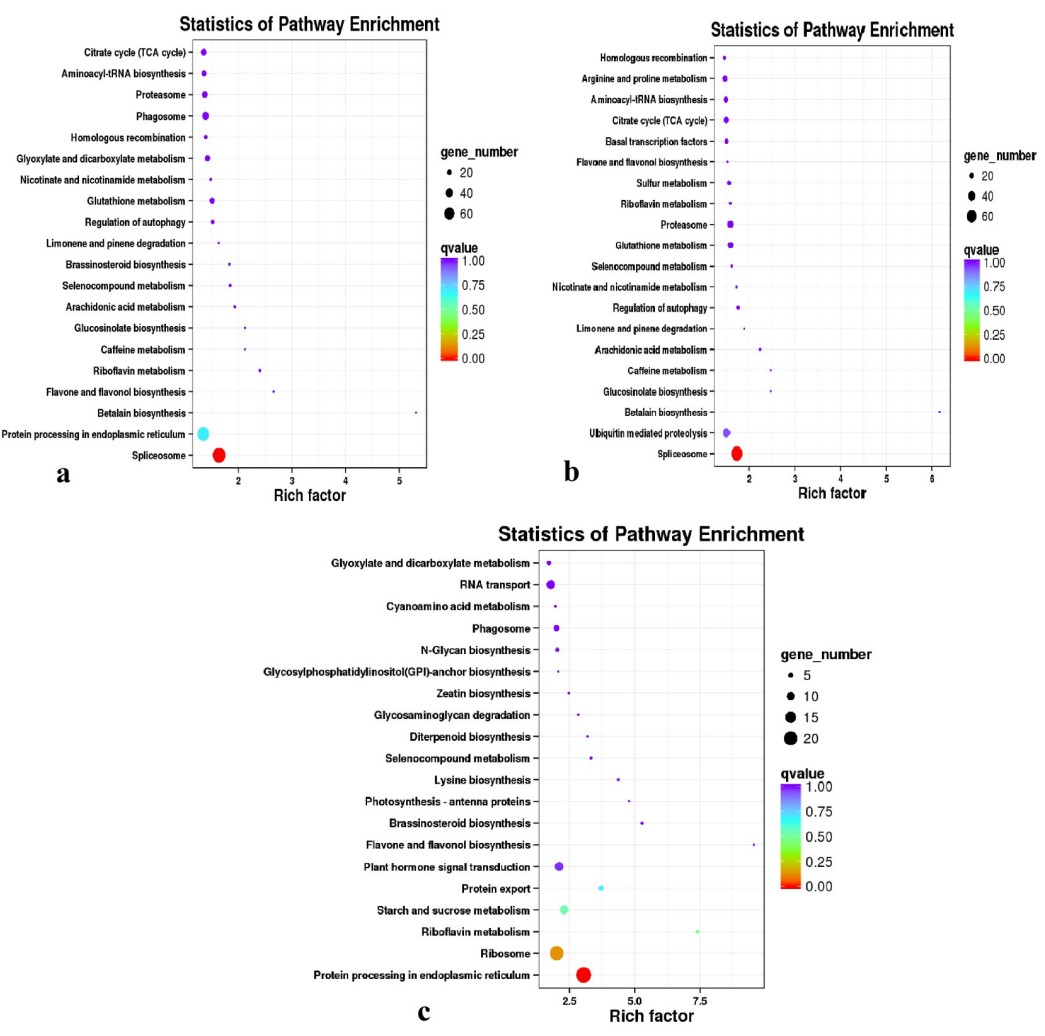

**Fig 9. Enrichment of KEGG pathways with genes that were differentially expressed between Wenzhou (WZ) and Haikou (HK) rhizomes.** Pathway name and enrichment intensity are shown in the right-hand legend. The Q value is the corrected *P*-value (false discovery rate). The enrichment factor is the ratio of the number of differentially expressed genes (DEGs) in a pathway to the number of all genes in that pathway. (a, all DEGs; b, up-regulated DEGs; c, down-regulated DEGs).

oils and the content of curcumin, polysaccharide, and starch in the rhizomes of *C. wenyujin* sourced from the traditional production areas in Wenzhou and the more recently established production areas in Haikou. Notably in this regard, we detected significantly higher content of the volatile oil terpene, curcumin, and polysaccharides in *C. wenyujin* rhizomes from Wenzhou compared with that in the rhizomes of plants cultivated in the Haikou area, whereas the content of starch was significantly higher in the Haikou rhizomes. Additionally, we detected certain differences between Wenzhou and Haikou rhizomes with respect to the composition of the volatile oil, which is consistent with the observations of Huang et al., who found that the volatile oil content of *C. wenyujin* rhizomes from Haikou (2.9%) was slightly lower than that of rhizomes from Zhejiang (3.0%), and also reported considerable differences in the volatile oil constituents of plants growing in different habitats [39]. Further studies have also shown differences in the chemical constituents of volatile oils in *C. wenyujin* from different planting areas, which were found to be associated with differences in the quality of the corresponding

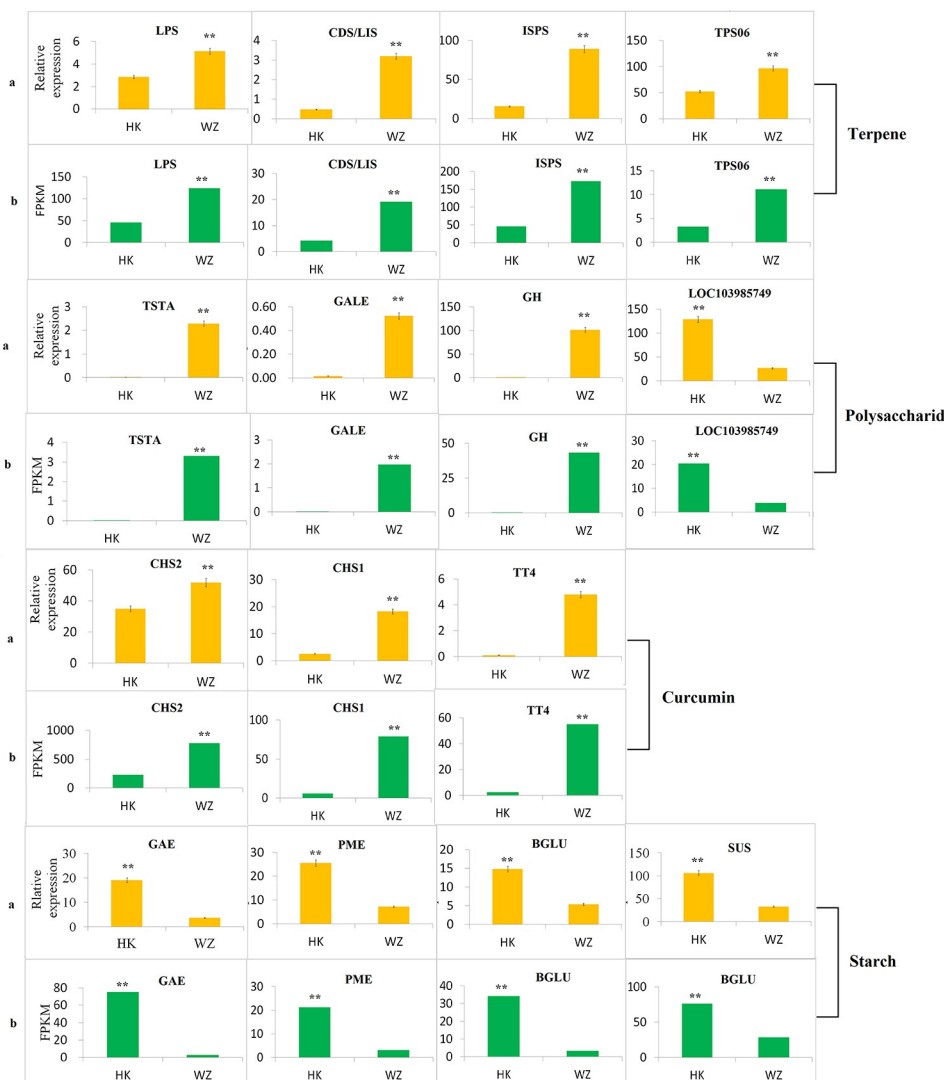

**Fig 10. qRT-PCR verification of genes that were differentially expressed between Wenzhou (WZ) and Haikou (HK) rhizomes (a) and the corresponding verification based on RNA-seq (b).** Using c121677 as a control/reference gene, the relative expression of genes was calculated using the $2^{-\Delta\Delta Ct}$ method. Asterisks denote significant differences (*, $P < 0.05$; **, $P < 0.01$; ***, $P < 0.001$). Error bars represent the standard deviations of mean values (n = 6). (CHS2: chalcone synthase2; CHS1: chalcone synthase1; TT4: chalcone and stilbene synthase family protein; LPS: levopimaradiene synthase; CDS/LIS: 3S-linalool synthase; ISPS: isoprene synthase; TPS06: dolabella-3,7-dien-18-ol synthase/sesquiterpene synthase 6; GAE: UDP-glucuronate 4-epimerase; PME: pectin methylesterase; BGLU: beta-glucosidase; SUS: sucrose synthase; TSTA: GDP-L-fucose synthase; GALE: UDP-glucose 4-epimerase; GH: glycoside hydrolase; LOC103985749: uncharacterized protein LOC103985749).

medicinal materials [17, 62, 63]. It has similarly been demonstrated that volatile oil content may vary depending upon the origin of the variety, cultivation area, cultivation techniques, and extraction method [8], and Cao et al. have reported that the content of volatile oil and curcumin differed substantially in different experimental areas, with the curcumin content varying from 1.5% to 5% and volatile oil content ranging from 0.4% to 0.7% [63]. In the present study, RT-PCR and transcriptome RNA-seq analyses revealed that genes involved in terpene, curcumin, and polysaccharide production were significantly up-regulated in *C. wenyujin* rhizomes from Wenzhou, thereby indicating that these genes are closely associated with observed

differences in the profiles of these metabolites in *C. wenyujin* cultivated in different areas. In contrast, we observed that starch-related pathways were significantly up-regulated in the rhizomes of *C. wenyujin* cultivated in Haikou. These results are consistent with the findings of previous studies that have indicated that *C. wenyujin* rhizomes from Wenzhou have higher content of the volatile oil, terpene, curcumin, polysaccharide but lower starch content than those in *C. wenyujin* rhizomes from Haikou. We also found that the gene expression patterns were consistent with those determined based on transcriptome FPKM analysis (Figs 3 and 10). Huang et al. have previously suggested that whereas differences in the starch content of *C. wenyujin* rhizomes from different producing areas are not appreciably influenced by latitude and longitude, they may be related to local soil, temperature, and light conditions, as well cultivation measures [64]. Starch formation is a photosynthesis-related dark reaction involving a series of enzymatic reactions within the chloroplast matrix, which is influenced to varying degrees by temperature, $CO_2$ concentration, and pH. It can therefore be speculated that differences in these environmental factors between Wenzhou and Haikou contribute to the observed differences in the content of starch and/or other substances in *C. wenyujin* rhizomes. Similarly, gene expression analysis has revealed that the curcumin content of *C. wenyujin* rhizomes is markedly affected by environmental factors and changes in nutritional conditions [19].

Transcriptomics is concerned with analysis of the gene expression levels and diversity profiles of different tissues in animals and plants [65–67], one of the important focuses of which is the study of the biosynthesis of metabolites in non-model medicinal plants [68, 69]. In the present study, we adopted hierarchical clustering to obtain complete transcriptional profiles of *C. wenyujin* cultivated in the Wenzhou and Haikou production areas, and accordingly found that the rhizomes of *C. wenyujin* grown in Wenzhou showed significant differences with respect to the transcriptional profiles of the co-expressed transcripts when compared with the rhizomes of plants grown in Haikou. Among these co-expressed transcripts, we found that when compared with those in Haikou rhizomes, the number of genes up-regulated in Wenzhou rhizomes was greater than those down-regulated. Functional analysis indicated that the enriched KEGG pathways appeared to be primarily associated with endocytosis and phagosome among cellular processes; ribosome, endoplasmic reticulum protein processing, RNA transport and spliceosome in genetic information processing; and carbon metabolism and biosynthesis of amino acids with respect to metabolism, thereby implying that the associated regulatory genes are significantly differentially expressed between the *C. wenyujin* rhizomes obtained from Wenzhou and Haikou.

Overall, we found that 980 DEGs were allocated to KEGG pathways, among which those associated with spliceosome, endoplasmic reticulum protein processing, ribosome, ubiquitin-mediated proteolysis, RNA transport, phagosome, glyoxylate and dicarboxylate metabolism, and glutathione metabolism were predominantly enriched. With respect to up-regulated DEGs, pathways associated with spliceosome, carbon metabolism, ubiquitin-mediated proteolysis, proteasome, citrate cycle (TCA cycle), aminoacyl-tRNA biosynthesis, arginine and proline metabolism, and glutathione metabolism were predominantly enriched, whereas for the down-regulated DEGs, pathways relating to ribosome, starch and sucrose metabolism, endoplasmic reticulum protein processing, plant hormone signal transduction, protein export, and riboflavin metabolism were significantly enriched. These findings are thus indicative of the active biosynthesis of not only secondary but also primary metabolites. Conversely, we noted that plant hormone signal transduction and starch and sucrose metabolism were inhibited and down-regulated in the rhizomes of *C. wenyujin* cultivated in Wenzhou. These DEGs annotations will provide a valuable resource for characterizing the relative importance of different signal transduction pathways and specific genes in the rhizomes of *C. wenyujin* grown in both

the traditional and recently established production areas [52], which in turn could provide useful information for examining biosynthesis mechanisms of interest in *C. wenyujin* (S4–S7 Figs).

In the present study, we found that among all genes that were differentially expressed between Wenzhou and Haikou rhizomes, the GO terms of plastid and chloroplast (CC); methylerythritol 4-phosphate (MEP) pathway, isopentenyl diphosphate biosynthetic process, and thylakoid membrane organization (BP), and protein kinase activity and protein serine/threonine kinase activity (MF) were predominantly enriched in the rhizomes from Wenzhou compared with those from Haikou. Among up-regulated DEGs, the GO terms of pollen tube tip (CC), leaf morphogenesis (BP), and polyamine oxidase activity, isomerase activity, ion channel activity, and phospholipase activity (MF) were significantly enriched in Wenzhou rhizomes, whereas for down-regulated DEGs, the GO terms cell wall (CC), defense response to fungus (BP), and peroxidase activity, copper ion binding, sucrose synthase activity, heat shock protein binding, and catalytic activity (MF) were significantly enriched in Wenzhou rhizomes. We assume that these differences reflect the fact that when grown under different environmental conditions, *C. wenyujin* requires different metabolites and levels of energy to respond to the different stimuli associated with the prevailing conditions.

When exposed to abiotic stresses, such as drought and elevated temperatures, plants respond by expressing a range transcription factors, induced via a series of signaling pathways. The stimulated transcription factors bind to corresponding *cis*-acting elements to initiate specific patterns of gene expression to counter the adverse effects of abiotic stress. Plant transcription factors known to respond to abiotic stress mainly include those in the *AP2/EREBP*, *MYB*, *WRKY*, *bZIP*, and *HSFs* [70–75]. *AP2/EREBP*, *MYB*, *WRKY*, *bZIP*, *HSFs*, *RLK*, *bHLH*, *NAC*, and *CAMK* families, the regulation of which is determined by a diverse range of biotic and abiotic pressures. plant growth and material metabolic processes, including trichome and seed coat development, embryogenesis leaf senescence, biosynthesis pathways, and the regulation of hormonal signals [76–80]. *C2H2* and *C2HC* zinc finger structures are characteristic features of the *WRKY* family of transcription factor from which the *WRKY* family can be explained [73]. In the present study, we detected significantly up-regulated expression of transcription factors in the *RLK*, *C2H2*, *Zn-clus*, *TKL*, *bZIP*, *CAMK*, *AP2/ERF*, *AGC*, *STE*, *C2C2*, *CMGC*, *C3H*, *SNF2*, *WRKY*, *bHLH*, *GNAT*, *MYB*, *HSF*, and *NAC* families. Thus, on the basis of our findings that the genes of transcription factors known to respond to abiotic environmental factors were significantly up-regulated, we can speculate that observed differences in the compositions and gene regulation of *C. wenyujin* rhizomes derived from the traditional (Wenzhou) and recently established (Haikou) production areas, are probably attributable to differences in environmental factors specific to these two regions.

## Conclusions

In summary, our findings indicate that the content of volatile oil, curcumin, and polysaccharides in the rhizomes of *C. wenyujin* cultivated in the Wenzhou production region is higher than those in the rhizomes of *C. wenyujin* grown in Haikou, whereas the content of starch is lower. We also detected significant differences in the constituents of the volatile oils of rhizomes obtained from the two production areas. Transcriptome analysis revealed notable differences in the gene expression patterns of rhizomes sourced from the Wenzhou and Haikou regions. We found that the expression of genes in pathways related to volatile oil, curcumin, and polysaccharide was significantly up-regulated in Wenzhou rhizomes, whereas that of starch-associated genes was significantly down-regulated, compared with those in Haikou rhizomes. Moreover, we identified the significantly up-regulated expression of transcription

factors (including those in the *RLK*, *C2H2*, *bZIP*, *CAMK*, *AP2/ERF*, *WRKY*, *bHLH*, *MYB*, *HSF*, and *NAC* families) that are typically associated with responses to abiotic environmental stress. Collectively, the findings of this study provide important insights into the molecular mechanisms underlying differences in the metabolite profiles of *C. wenyujin*, which are assumed to reflect differences in environmental factor characterizing the two examined production regions. We also describe new methodologies that may prove beneficial with respect to evaluating the authenticity of Chinese herbal medicines.

## Supporting information

**S1 Fig. *Curcuma* unigene length distribution.**
(TIF)

**S2 Fig. Comparison of RNA-seq and quantitative reverse-transcription PCR (qRT-PCR) analyses of 15 selected genes.** log2Fold Change [Wenzhou (WZ)/Haikou (HK)].
(TIF)

**S3 Fig. Transcription factor expression for Wenzhou (WZ) versus Haikou (HK).**
(TIF)

**S4 Fig. Terpenoid backbone synthesis: KEGG enrichment structure for Wenzhou (WZ) versus Haikou (HK).**
(TIF)

**S5 Fig. Flavonoid biosynthesis: KEGG enrichment structure for Wenzhou (WZ) versus Haikou (HK).**
(TIF)

**S6 Fig. Amino sugar and nucleotide sugar metabolism: KEGG enrichment structure for Wenzhou (WZ) versus Haikou (HK).**
(TIF)

**S7 Fig. Starch and sucrose metabolism: KEGG enrichment structure for Wenzhou (WZ) versus Haikou (HK).**
(TIF)

**S1 Table. Environmental and weather conditions in Wenzhou and Haikou.**
(XLS)

**S2 Table. RNA-seq verification using RT-PCR primers in *Curcuma wenyujin*.**
(XLS)

**S3 Table. GC-MS detection results of the main volatile oil constituents in *Curcuma wenyujin* from Wenzhou (WZ) and Haikou (HK).**
(XLS)

**S4 Table. Sample sequencing data evaluation statistics.**
(XLS)

**S5 Table. Assembly result statistics.**
(XLS)

**S6 Table. Comparison of sequencing data and assembly results.**
(XLS)

**S7 Table. Unigene annotation statistics.**
(XLS)

**S8 Table. Expression data of enriched key DEGs in different assemblies for Wenzhou (WZ) versus Haikou (HK).**
(XLS)

**S9 Table. Details of topGO enrichment for Wenzhou (WZ) versus Haikou (HK).**
(XLS)

**S10 Table. Details of KEGG enrichment for Wenzhou (WZ) versus Haikou (HK).**
(XLS)

**S11 Table. Transcription factor expression for Wenzhou (WZ) versus Haikou (HK).**
(XLS)

**S1 File. Enrichment analysis of responsive genes and transcripts using GO terms for Wenzhou (WZ) versus Haikou (HK).** The node size is proportional to the number of targets in the GO category. Node color represents enriched significance; a deeper color represents a higher significance [a1, a2, and a3: total (topGO_BP, topGO_CC, and topGO_MF); b1, b2, and b3: down-regulated (topGO_BP, topGO_CC, and topGO_MF); c1, c2, and c3: up-regulated (topGO_BP, topGO_CC, and topGO_MF)].
(ZIP)

## Acknowledgments

We would like to thank Editage (www.editage.cn) for editing the English language in this manuscript.

## Author Contributions

**Conceptualization:** Lilan Lu.

**Data curation:** Lilan Lu, Peiwei Liu.

**Supervision:** Yuxiu Zhang, Caixia Wang, Jian Feng.

**Validation:** Peiwei Liu.

**Writing – original draft:** Lilan Lu.

**Writing – review & editing:** Yanfang Yang, Jianhe Wei.

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
