## [Decision Letter · Decision Letter 0]

4 Dec 2019

PONE-D-19-24237

Transcriptome analysis of Curcumawenyujin from Haikou and Wenzhou and a comparison of the main substances and related genes of Rhizoma Curcumae

PLOS ONE

Dear Dr. Lilan,

Thank you for submitting your manuscript to PLOS ONE. After careful consideration, we feel that it has merit but does not fully meet PLOS ONE’s publication criteria as it currently stands. Therefore, we invite you to submit a revised version of the manuscript that addresses the points raised during the review process.

Both reviewers pointed out deficiencies in description of methodological details, conditions and sample details and conditions. According to Publication Criterion #3 You are required to describe your methods and samples in sufficient detail to ensure replicability. This extends also to purchased products that may not be sufficiently described or characterized, or whose purchase locations were not provided.

You are required to state how the Illumina RNASeq was done (details and instrument) and submit the reads to NCBI SRA Archive (PloS ONE publication criterion #7 Data Availability (see also https://journals.plos.org/plosone/s/data-availability).

We would appreciate receiving your revised manuscript by Jan 18 2020 11:59PM. To enhance the reproducibility of your results, we recommend that if applicable you deposit your laboratory protocols in protocols.io, where a protocol can be assigned its own identifier (DOI) such that it can be cited independently in the future. For instructions see: http://journals.plos.org/plosone/s/submission-guidelines#loc-laboratory-protocols

We look forward to receiving your revised manuscript.

Kind regards,

Christian Schönbach, Dr.rer.nat.

Academic Editor

PLOS ONE

Journal Requirements:

Reviewers' comments:

Reviewer's Responses to Questions

**Comments to the Author**

1. Is the manuscript technically sound, and do the data support the conclusions?

Reviewer #1: Partly

Reviewer #2: Yes

2. Has the statistical analysis been performed appropriately and rigorously? 

Reviewer #1: Yes

Reviewer #2: Yes

3. Have the authors made all data underlying the findings in their manuscript fully available?

Reviewer #1: Yes

Reviewer #2: Yes

4. Is the manuscript presented in an intelligible fashion and written in standard English?

Reviewer #1: Yes

Reviewer #2: No

5. Review Comments to the Author

Reviewer #1: Curcumae Rhizoma, known as Ezhu (Chinese), and Curcumae Radix, known as Yujin (Chinese), are different plant parts coming from three same species of Curcuma according to China Pharmacopoeia. The Chinese Pharmacopoeia recorded that Curcumae Radix should be the dry radix of Curcuma wenyujin Y. H. Chen and C. Ling, C. longa L., C. kwangsiensis S. G. Lee, and C. phaeocaulis Valeton. And Curcumae Rhizoma should be the dry rhizomes derived from the above-mentioned species except C. longa L. They are similar in source but different in medicinal parts. Curcumae Rhizoma and Curcumae Radix are confused in variety and source, even in clinical trials by some nonprofessional workers. So, it is important for us to make them clear. Also, genotype and environment interaction is high for secondary metabolites in Curcuma sp. The main objective of the present study was to analyze the constituents of essential oils such as terpene, curcumin, polysaccharide, starch, and other important substances from C. Wenyujin rhizomes from the traditional (Wenzhou) and introduced (Haikou) production areas. In this study authors also attempted comparative RNAseq profiles of C. Wenyujin rhizomes produced in traditional (Wenzhou) and introduced (Haikou) areas. The study has some interesting findings in the poorly understood crop species of Curcuma.

However, authors need to clarify following points.

The introduction part seems to be too lengthy – cab be shortened.

L.71: The terminology 'Species variety' is not botanically correct. Authors may use the term ‘genotype’ instead.

L.86: There is nothing called “lime soil”. Is it alkaline soil?

L.120-122: Give a suitable reference for this statement.

L.146: What do you mean by treatment conditions?

L.146-153: Authors silent on experimental conditions. Similar sampling and growing conditions are important for studying secondary metabolites and comparative transcriptomics.

The environmental and weather conditions of the two provinces need to be given as supplementary table to understand the differences in secondary metabolites and related genes.

L.154: Protocol for RNA isolation is not mentioned. Also specify the tissue from which RNA was isolated.

In case of RNASeq, number of biological replicates sequences is not clear. Also, replicates need to be compared at sequence level and prove that they are not significantly different.

L.209: Sufficient experimental details viz., temperature, pressure and slice thickness need to be given on boiling and drying of samples?

L.234: What is the sample? rhizome or leaf, or at what stage?

L. 248: Is it a curcumin or curcuminoids

L.268: Polysaccharide and starch content methodology are too elaborate and need to shorten with suitable references.

L 361………..

In many literatures, the C. weeyujin is referred as synonymous to C. aromatica. In that case can authors compare the present transcritomes with already available C. aromatica transcriptome to draw the meaningful conclusions.

Hope above suggestions help authors to improve the manuscript.

Reviewer #2: Language correction is essential before acceptance. Method used for RNA isolation is not mentioned (what mentioned as RNAse free DNAse, Takara; is just an enzyme and is not the method used for RNA extraction).

Minor Corrections Suggested

1) Line no. 38, 39: better replace ‘was’ with were.

2) Species name should always be in small letters, throughout the manuscript its written as CurcumaWenyujin , few line numbers are mentioned ( Line no.119, 134, 148, 342, 350, 347, 355, 506, 509, 522, 528, 538, 541, 544, 546, 549, 558, 562, 563, 574, 582, 583, 594, 609, 611, 625, 627, 644, 650).

3) Make sure that Curcuma wenyujin is in italics (not followed).

4) Line no. 265: Rephrase the sentence in a better conveyable form.

5) Line no. 297: spelling correction; ‘using’ is written as uing.

6) Line no. 329: better replace ‘was’ with were.

7) Line no. 561 is contradictory to Line no. 564 (about polysaccharide).

6. PLOS authors have the option to publish the peer review history of their article (what does this mean?). If published, this will include your full peer review and any attached files.

Reviewer #1: No

Reviewer #2: No

---

## [Author Response · Author response to Decision Letter 0]

29 Feb 2020

Dear Plos one editor and reviewers,

We thank you for all of your comments and suggestions about ways to improve our manuscript, now retitled “Transcriptome analysis of Curcuma wenyujin from Haikou and Wenzhou and a comparison of the main constituents and related genes of Rhizoma Curcumae”. The following are our response to your questions and suggestions.

Revision: Thanks for the suggestion, We have revised the manuscript to meet PLOS ONE's style requirements by consulting The PLOS ONE style templates.

Revision: Thanks for the suggestion, We have created a new ORCID：0000-0003-4317-0029

3. Both reviewers pointed out deficiencies in description of methodological details, conditions and sample details and conditions. According to Publication Criterion #3 You are required to describe your methods and samples in sufficient detail to ensure replicability. This extends also to purchased products that may not be sufficiently described or characterized, or whose purchase locations were not provided.

Revision: Thanks for the suggestion, I read it carefully and have made changes in the text. The purchased products was sufficiently described or characterized, purchase locations were provided in the paper. We have described methods and samples in sufficient detail to ensure replicability in the revised manuscript.

4. You are required to state how the Illumina RNASeq was done (details and instrument) and submit the reads to NCBI SRA Archive (PloS ONE publication criterion #7 Data Availability (see also https://journals.plos.org/plosone/s/data-availability).

Revision: Thanks for the suggestion, the data reads of Illumina sequencing have been deposited in the National Center for Biotechnology Information Sequence Read Archive (SRA accession: PRJNA598542; Temporary Submission ID: SUB6764411; SAMPLE: Curcuma wenyujin (SAMN13707351); Release date: 2020-01-08. https://www.ncbi.nlm.nih.gov/sra/PRJNA598542).

Additional Editor Comments:

1. Is the manuscript technically sound, and do the data support the conclusions? The manuscript must describe a technically sound piece of scientific research with data that supports the conclusions. Experiments must have been conducted rigorously, with appropriate controls, replication, and sample sizes. The conclusions must be drawn appropriately based on the data presented. 

Reviewer #1: Partly

Reviewer #2: Yes

Revision: Thanks for the suggestion, According to this suggestion “Experiments must have been conducted rigorously, with appropriate controls, replication, and sample sizes. The conclusions must be drawn appropriately based on the data presented. ” We have Modified this article based on detailed recommendations from experts in red highlight place.

 2. Is the manuscript presented in an intelligible fashion and written in standard English? PLOS ONE does not copyedit accepted manuscripts, so the language in submitted articles must be clear, correct, and unambiguous. Any typographical or grammatical errors should be corrected at revision, so please note any specific errors here.

Reviewer #1: Yes

Reviewer #2: No

Revision: Thanks for the suggestion, This language polish has been corrected by the company(Editage) polish, to ensure clear, correct, and unambiguous.

Review Comments to the Author

Reviewer #1”: The introduction part seems to be too lengthy – cab be shortened.

Revision: Thanks for the suggestion, We have shortened the introduction in the revised manuscript.

Reviewer #1: L.71: The terminology 'Species variety' is not botanically correct. Authors may use the term ‘genotype’ instead.

Revision: Line 71: “Species variety” was replaced with “genotype”. Shown in line 69 in the revised manuscript. 

Reviewer #1: L.86: There is nothing called “lime soil”. Is it alkaline soil?

Revision: Line 86: “lime soil” was replaced with “alkaline soil”. Shown in line83 in the revised manuscript. 

Reviewer #1: L.120-122: Give a suitable reference for this statement.

Revision: Thanks for the suggestion, we have given a suitable reference for this statement in line 104 in the revised manuscript.

Reviewer #1: L.146: What do you mean by treatment conditions?

Revision: Thanks for the suggestion, L.146: “Plant materials and treatment conditions? ” was replaced with “Plant materials”. Shown in line 131in the revised manuscript. 

Reviewer #1: L.146-153: Authors silent on experimental conditions. Similar sampling and growing conditions are important for studying secondary metabolites and comparative transcriptomics. The environmental and weather conditions of the two provinces need to be given as supplementary table to understand the differences in secondary metabolites and related genes.

Revision: Thank you for the recommends. The environmental and weather conditions of the two provinces was given in S1 Table (shown in L141 in the revised manuscript).

Reviewer #1: L.154: Protocol for RNA isolation is not mentioned. Also specify the tissue from which RNA was isolated. In case of RNASeq, number of biological replicates sequences is not clear. Also, replicates need to be compared at sequence level and prove that they are not significantly different.

Revision: L.154: “Protocol for RNA isolation and specify the tissue from which RNA was isolated ”is added in L143-145.Three biological replicates samples in sequences were mentioned, and were shown in line 145 in the revised manuscript. “replicates need to be compared at sequence level and prove that they are not significantly different” were mentioned in line 324-326 in the revised manuscript. 

Reviewer #1: L.209: Sufficient experimental details viz., temperature, pressure and slice thickness need to be given on boiling and drying of samples?

Revision: L.209: “ Sufficient experimental details viz., temperature, pressure and slice thickness need to be given on boiling and drying of samples? ” were revised . Shown in line 205-207in the revised manuscript.

Reviewer #1: L.234: What is the sample? rhizome or leaf, or at what stage?

Revision: L.234: “What is the sample? rhizome or leaf, or at what stage? ” were revised with “C. wenyujin rhizomes ” . Shown in line 230 in the revised manuscript.

Reviewer #1: L. 248: Is it a curcumin or curcuminoids

Revision: Curcuminoids mainly cantain curcumin , demethoxycurcumin, bisdemethoxycurcumin and other curcumin substances, and in this paper, we analyzed a curcumin in rhizome of Curcuma wenyujin, so, "It is a curcumin in this paper" was confirmed, shown in line L244 in the revised manuscript.

Reviewer #1: L.268: Polysaccharide and starch content methodology are too elaborate and need to shorten with suitable references.

Revision: Thank you for the recommends. L.268: “Polysaccharide and starch content methodology are too elaborate and need to shorten with suitable references” was revised and shorten with suitable references[53-54]. Shown in line L264-279 in the revised manuscript.

Reviewer #1: In many literatures, the C. weeyujin is referred as synonymous to C. aromatica. In that case can authors compare the present transcritomes with already available C. aromatica transcriptome to draw the meaningful conclusions. Hope above suggestions help authors to improve the manuscript.

Revision: Thank you for the recommends. In discussions, We cited the literature and discussed and compared C. wenyujin between HK and WZ areas. Shown in line L486-489 in the revised manuscript.

Reviewer #2: Language correction is essential before acceptance. Method used for RNA isolation is not mentioned (what mentioned as RNAse free DNAse, Takara; is just an enzyme and is not the method used for RNA extraction).

Revision: This language polish has been corrected by the company (Editage) polish, and RNA extraction methods have been mentioned in the article L43-145.

Reviewer #2: Line no. 38, 39: better replace ‘was’ with were.

Revision: Line 38, 39: “was” was replaced with “were”. Shown in line 39 in the revised manuscript. 

Reviewer #2: Species name should always be in small letters, throughout the manuscript its written as Curcuma Wenyujin, few line numbers are mentioned ( Line no.119, 134, 148, 342, 350, 347, 355, 506, 509, 522, 528, 538, 541, 544, 546, 549, 558, 562, 563, 574, 582, 583, 594, 609, 611, 625, 627, 644, 650).

Revision: Line (119, 134, 148, 342, 350, 347, 355, 506, 509, 522, 528, 538, 541, 544, 546, 549, 558, 562, 563, 574, 582, 583, 594, 609, 611, 625, 627, 644, 650) “Species name should always be in small letters, throughout the manuscript its written as Curcuma Wenyujin ” were revised. Shown in Line (100, 111, 134, 289, 295, 297, 302, 457, 461, 474, 480, 491, 493, 497, 498, 501, 510, 512, 513, 515, 524, 532, 533, 544, 559, 561, 574, 577, 594) in the revised manuscript.

Reviewer #2: Make sure that Curcuma wenyujin is in italics (not followed).

Revision: “Curcuma wenyujin is in italics ” were revised . Shown in the revised manuscript.

Reviewer #2: Line no. 265: Rephrase the sentence in a better conveyable form.

Revision: Line no. 265“Rephrase the sentence in a better conveyable form”were revised . Shown in line 260-265 in the revised manuscript.

Reviewer #2: Line no. 297: spelling correction; ‘using’ is written as uing.

Revision: Line no. 297 “uing” was replaced with “using”. Shown in line 267 in the revised manuscript. 

Reviewer #2: Line no. 329: better replace ‘was’ with were.

Revision: Line no. 329 was revised. Shown in line 274-279 in the revised manuscript.

Reviewer #2: Line no. 561 is contradictory to Line no. 564 (about polysaccharide).

Revision: Line no561-564 “561 is contradictory to Line no. 564 (about polysaccharide) in Line no.561-564” were revised . Shown in line 511-515 in the revised manuscript.

Other revisions: 

1. Line 469 “15 of the transcriptomes of differentially expressed genes- How did you select these genes? What was the criteria followed to shortlist 15 genes? ” 

Revision: These 15 genes are genes related to terpenoids, curcumin, polysaccharides, and starch metabolism pathways of Rhizoma Curcuma, and the selected genes are expressed in both samples. The differential expression of genes in the two regions is high, and the p value and FDR value are within the range. These genes are designed with appropriate primers and quantified.

2. Line 376“A total of six samples-what are they? ”

Revision: Thank you for the recommends. they are HK1, HK2, HK3, WZ1, WZ2, WZ3. Shown in line 326-327 in the revised manuscript.

Best regards,

Lulilan,

---

## [Decision Letter · Decision Letter 1]

8 Apr 2020

PONE-D-19-24237R1

Transcriptome analysis of Curcuma wenyujin from Haikou and Wenzhou and a comparison of the main constituents and related genes of Rhizoma Curcumae

PLOS ONE

Dear Dr. Lu Lilan,

Thank you for submitting your manuscript to PLOS ONE. After careful consideration, we feel that it has merit but does not fully meet PLOS ONE’s publication criteria as it currently stands. Therefore, we invite you to submit a revised version of the manuscript that addresses the points raised during the review process.

While some of the major issuses raised by reviewers in the previous round were addressed more work is required improve descriptions of methods and language to meet publicaiton criteria #3 and #5. Especially, add parameters and thresholds to methods applied, explain rational for chosing a particular statistical test (here univariate ANOVA), and most importantly revisit and revise the DEG analysis part to ensure enrichment results are statically correctly supported. In additon your are required to improve lablelling and quality of figures.

We would appreciate receiving your revised manuscript by May 23 2020 11:59PM. To enhance the reproducibility of your results, we recommend that if applicable you deposit your laboratory protocols in protocols.io, where a protocol can be assigned its own identifier (DOI) such that it can be cited independently in the future. For instructions see: http://journals.plos.org/plosone/s/submission-guidelines#loc-laboratory-protocols

We look forward to receiving your revised manuscript.

Kind regards,

Christian Schönbach, Dr.rer.nat.

Academic Editor

PLOS ONE

Reviewers' comments:

Reviewer's Responses to Questions

**Comments to the Author**

1. If the authors have adequately addressed your comments raised in a previous round of review and you feel that this manuscript is now acceptable for publication, you may indicate that here to bypass the “Comments to the Author” section, enter your conflict of interest statement in the “Confidential to Editor” section, and submit your "Accept" recommendation.

Reviewer #2: All comments have been addressed

Reviewer #3: (No Response)

2. Is the manuscript technically sound, and do the data support the conclusions?

Reviewer #2: Yes

Reviewer #3: Partly

3. Has the statistical analysis been performed appropriately and rigorously? 

Reviewer #2: Yes

Reviewer #3: No

4. Have the authors made all data underlying the findings in their manuscript fully available?

Reviewer #2: Yes

Reviewer #3: Yes

5. Is the manuscript presented in an intelligible fashion and written in standard English?

Reviewer #2: Yes

Reviewer #3: No

6. Review Comments to the Author

Reviewer #2: Authors incorporated the suggestions into their manuscript and as a result the MS quality has been improved. The revised MS can be accepted as it is.

Reviewer #3: The authors analyzed C. wenyujin rhizomes in two different locations and showed that the difference in metabolome and transcriptome were consistent.

The conclusion sounds OK but there are so many points to be improved in figures, statistical analysis and sentences.

The comparison along the metabolic pathway, and their conclusion would be OK. However, the distribution of transcriptome should be evaluated and normalized before detail statistical analysis such as DEG. Also, the detail about clustering analysis and enrichment analysis are not clear to evaluate them.

All of the figures were looks very low in resolution (converted from compressed JPEG?) and most of the labels are difficult to read, even I downloaded each tiff file.

Use vector format images or, at least, more high resolution and low compression images.

It is strongly recommended to submit the whole manuscript to an English editorial service.

There are detail comments below.

l85. ~ indicating that the oil content of Curcuma rhizomes can

85 be affected by factors such as soil quality and climate.

> Add some reference if there are some preceding related studies.

l. 192

> It would be specify the name of the test and parameters since "DESeq" is a name of a function in the R package rather than a name of a method.

l. 282 Univariate analysis of variance was applied to determine the significance of the results among different treatments.

> The target of the analysis unclear. Does it means "different species", and the variables are compound components? Since they have only two groups, are there any rational reason to apply univariate ANOVA instead of t-test or U-test?

Through three replications, the average value of each parameter was calculated, and the standard error (SE) of the mean value was obtained. Univariate analysis of variance was applied to determine the significance of the results among different treatments. Multidimensional tests were considered significant at a P value < 0.05. SPSS V.13 was used for statistical analysis.

l 327 41.74 Gb

> If it means giga "byte", show the amount in giga "base pair" instead.

l. 328 Q30 percentages

> It should be more concretely and use general term.

For example, "Number of samples which Phred quality score is higher than 30."

l. 340 1.00e-05

> It would be better to use scientific format (superscripts) instead of computational format.

l 340

> In the DEG analysis the number of up regulated genes were much larger than that of down regulated. The authors should concern about experimental bias between samples or consider to apply some compensation such as quantile normalization.

Fig 4 shows the ratio of DEG unigenes in each ontology is highly correlated

with that of all annotated unigenes.

> It seems that DEG are not correlate with their functions but almost randomly sampled from the all unigenes, so that the significance of enrichment analysis is rather doughtful.

Fig. 7

> Make clear the number of genes shown in this figure, is it 3720? If it is so make clear how and why these genes were chosen. It is not clear what means "up/down regulated" in this figure, I mean compared with what values.

> And the labels in the columns should be replaced by their real labels (T01 = HK1, T02 = HK2, T03 = HK3, T04 = WZ1, T05 = 393 WZ2, T06 = WZ3).

l. 409 A comparison of DEGs in 26 COG classifications between HK and WZ is shown in Fig 8.

> It is not clear that the numbers of DEG in this figure means up regulated or down regulated. And for enrichment analysis, it is also important to show the number of annotated genes in each category and the ratio of DEG to the annotated genes.

l. 454

> It seems meaningless to show the sequence id such like c118842.grraph-co, show gene name or protein name instead. It would be much better to show them in the pathway map also.

Fig. 9

> Arrange the plots for FPKM and rt-PCR side by side when the comparison about the same gene is important.

7. PLOS authors have the option to publish the peer review history of their article (what does this mean?). If published, this will include your full peer review and any attached files.

Reviewer #2: No

Reviewer #3: Yes: Naoaki ONO

---

## [Author Response · Author response to Decision Letter 1]

6 Sep 2020

Dear Plos one editor and reviewers,

We thank you for all of your comments and suggestions about ways to improve our manuscript, and give us a second chance to revise the paper, now retitled “Transcriptome analysis of Curcuma wenyujin from Haikou and Wenzhou, and a comparison of the main constituents and related genes of Rhizoma Curcumae”. The following are our response to your questions and suggestions.

While some of the major issuses raised by reviewers in the previous round were addressed more work is required improve descriptions of methods and language to meet publicaiton criteria #3 and #5. Especially, add parameters and thresholds to methods applied, explain rational for chosing a particular statistical test (here univariate ANOVA), and most importantly revisit and revise the DEG analysis part to ensure enrichment results are statically correctly supported. In additon your are required to improve lablelling and quality of figures.

Revision: Thank you very much for the suggestion, and We have accepted the expert's suggestion that we improve the description of the method and language by polishing the company to meet the #3 and #5 public standards. In particular, Why are parameters and thresholds added to the applied method to explain the rationality of choosing a specific statistical test (here, univariate ANOVA), Curcuma rhizomes from different species, and they have only two groups, We had applied univariate ANOVA instead of t-test or U-test. Seen in lines 282 in the revised. and the DEG analysis part and enrichment results have been modified and explained. In addition, we also made adjustments to meaningful tags and improved image quality. We have Modified this article based on detailed recommendations from experts in red highlight place.

Comments to the Author

1. If the authors have adequately addressed your comments raised in a previous round of review and you feel that this manuscript is now acceptable for publication, you may indicate that here to bypass the “Comments to the Author” section, enter your conflict of interest statement in the “Confidential to Editor” section, and submit your "Accept" recommendation.

Reviewer #2: All comments have been addressed

Reviewer #3: (No Response)

2. Is the manuscript technically sound, and do the data support the conclusions?

Reviewer #2: Yes

Reviewer #3: Partly

Revision: Thanks for the suggestion, According to this suggestion “Experiments must have been conducted rigorously, with appropriate controls, replication, and sample sizes. The conclusions must be drawn appropriately based on the data presented. ” We have Modified this article based on detailed recommendations from experts in red highlight place.

3. Has the statistical analysis been performed appropriately and rigorously?

Reviewer #2: Yes

Reviewer #3: No

Revision: Thanks for the suggestion, According to this suggestion “the statistical analysis been performed appropriately and rigorously. ” and according to this suggestion We had applied univariate ANOVA instead of t-test or U-test. Seen in lines 282 in the revised manuscript. We have Modified this article based on detailed recommendations from experts in red highlight place.

4. Have the authors made all data underlying the findings in their manuscript fully available?

Reviewer #2: Yes

Reviewer #3: Yes

5. Is the manuscript presented in an intelligible fashion and written in standard English?

Reviewer #2: Yes

Reviewer #3: No

Revision: Thanks for the suggestion, This language polish has been corrected by the company(Editage) polish, to ensure clear, correct, and unambiguous. Seen in red highlight place in the revised manuscript.

6. Review Comments to the Author

Reviewer #2: Authors incorporated the suggestions into their manuscript and as a result the MS quality has been improved. The revised MS can be accepted as it is.

Answer: Thank you for your comments

Reviewer #3: 

The authors analyzed C. wenyujin rhizomes in two different locations and showed that the difference in metabolome and transcriptome were consistent. The conclusion sounds OK but there are so many points to be improved in figures, statistical analysis and sentences. The comparison along the metabolic pathway, and their conclusion would be OK. However, the distribution of transcriptome should be evaluated and normalized before detail statistical analysis such as DEG. Also, the detail about clustering analysis and enrichment analysis are not clear to evaluate them.

Revision: Thank you for your comments, the distribution of transcriptome was evaluated and normalized before detail statistical analysis such as DEG and unigenes in S1-Fig, S4-S7 Table. Also, the detail about clustering analysis and enrichment analysis are evaluate them in S9, S10 Table and S1 File. In-depth research in the future, we will further analyze.

All of the figures were looks very low in resolution (converted from compressed JPEG?) and most of the labels are difficult to read, even I downloaded each tiff file.Use vector format images or, at least, more high resolution and low compression images.

Revision: Thank you for your comments, all of the figures were adjusted by using more high resolution and low compression images.

It is strongly recommended to submit the whole manuscript to an English editorial service. 

Revision: Thank you for your comments, We had submited the whole manuscript to an English editorial service again (Editage) . Seen in red highlight place.

There are detail comments below.

L85. ~ indicating that the oil content of Curcuma rhizomes can be affected by factors such as soil quality and climate. Add some reference if there are some preceding related studies.

Revision: Thank you for your comments. We have added a reference about preceding related studies, However, now, There is still not much research on related studies about the soil quality and climate, in lines L84-88 in the revised manuscript.

L192: It would be specify the name of the test and parameters since "DESeq" is a name of a function in the R package rather than a name of a method.

Revision: Thank you for your comments. "DESeq" is revised for a name of a function in the R package. The P value results were regulated using the Benjamin and Hochberg method (1995) for controlling the false discovery rate (FDR), and genes at a P value < 0.05 were classified as differentially expressed genes (DEGs), in lines L198-202 in the revised manuscript.

L282 : Univariate analysis of variance was applied to determine the significance of the results among different treatments. The target of the analysis unclear. Does it means "different species", and the variables are compound components? Since they have only two groups, are there any rational reason to apply univariate ANOVA instead of t-test or U-test?

Revision: Thank you for your comments. Curcuma rhizomes from different species, and they have only two groups, We had applied univariate ANOVA instead of t-test or U-test. Seen in lines 288 in the revised manuscript.

L:327 41.74 Gb, If it means giga "byte", show the amount in giga "base pair" instead.

Revision: Thank you for your comments, it means giga "byte", 41.74 Gb was revised for 41.74 Gbp (base pair) in lines 335 in the revised manuscript.

L: 328 Q30 percentages, It should be more concretely and use general term.

For example, "Number of samples which Phred quality score is higher than 30."

Revision: Thank you for your comments, Q30 percentages was revised for general term that number of samples which Phred quality score is higher than 30) in lines 336-337 in the revised manuscript.

L 340: 1.00e-05, It would be better to use scientific format (superscripts) instead of computational format.

Revision: Thank you for your comments. It has been used scientific format (superscripts)( 1.00-5) instead of computational format in lines 342 in the revised manuscript.

L 340: In the DEG analysis the number of up regulated genes were much larger than that of down regulated. The authors should concern about experimental bias between samples or consider to apply some compensation such as quantile normalization.

Revision: Thank you for your comments. Yes, you are right about that “experimental bias between samples or consider to apply some compensation such as quantile normalization”. In the DEG analysis the number of up /down regulated genes, logarithm quantile normalization (Log2FC（WZ/HK）) were seen in S8 Table .

Fig 4 shows the ratio of DEG unigenes in each ontology is highly correlated with that of all annotated unigenes.It seems that DEG are not correlate with their functions but almost randomly sampled from the all unigenes, so that the significance of enrichment analysis is rather doughtful.

Revision: Thank you for your comments. Fig4 showed Gene ontology (GO) histogram of classification in annotated unigenes WZ (Wenzhou) versus HK (Haikou). The functions of the unique sequences extracted from C. wenyujin were classified based on the three GO terms: cellular component (CC), biological process (BP), and molecular function (MF), The DEGs were mainly assigned to ‘cell, go: 0005623’, ‘cell part, go: 0044464’, ‘organelle, go: 0043226’, ‘membrane, go: 0016020’,‘cellular process, go: 0009987’, ‘metabolic process, go: 0008152’, ‘single-organism process, go: 0044699’, ‘binding, go: 0005488’, and ‘catalytic activity, go: 0003824’ (Fig 4). and We had some KEEG and GO enrichment analysis seen in lines 405-418 and 446-458 in the revised manuscript and related enrichment information was seen in S9, S10 Table and S1 File. We will conduct more in-depth research in the future on the significance of enrichment analysis.

Fig. 7: Make clear the number of genes shown in this figure, is it 3720? If it is so make clear how and why these genes were chosen. It is not clear what means "up/down regulated" in this figure, I mean compared with what values.And the labels in the columns should be replaced by their real labels (T01 = HK1, T02 = HK2, T03 = HK3, T04 = WZ1, T05 = 393 WZ2, T06 = WZ3).

Revision: Thank you for your comments. Make clear the number of genes shown in this figure is 4620. These genes were chosen because of a significant difference in these genes between WZ and HK . the values of up/down regulated" in this figure were those of WZ compared HK. And the labels in the columns have been replaced by their real labels (HK1, HK2, HK3, WZ1, WZ2, WZ3) in Fig 7 in the revised manuscript.

L409: A comparison of DEGs in 26 COG classifications between HK and WZ is shown in Fig 8. It is not clear that the numbers of DEG in this figure means up regulated or down regulated. And for enrichment analysis, it is also important to show the number of annotated genes in each category and the ratio of DEG to the annotated genes.

Revision: Thank you for your comments, We mean that 26 COG functional classifications of consensus sequence of DEGs between HK and WZ are shown in Fig 8, not a comparison of DEGs in 26 COG classifications between HK and WZ is shown in Fig 8. We have revised it seen in Line 409. We have showed the number of annotated genes in main eight categories and the ratio of DEG to the annotated genes, seen in lines 424-432in the revised manuscript.

L 454:It seems meaningless to show the sequence id such like c118842.grraph-co, show gene name or protein name instead. It would be much better to show them in the pathway map also.

Revision: Thank you for your comments, We have showed gene name or protein name instead of the sequence id such like c118842.grraph-co seen in lines 475-478 in the revised manuscript.

Fig. 10Arrange the plots for FPKM and rt-PCR side by side when the comparison about the same gene is important.

Revision: Thank you for your comments, We have arranged the plots for FPKM (DGE) and RT-PCR side by side when the comparison, seen in Fig. 10 in the revised manuscript.

7. PLOS authors have the option to publish the peer review history of their article (what does this mean?). If published, this will include your full peer review and any attached files.Answer: Thank you, We choose “no”.

---

## [Decision Letter · Decision Letter 2]

10 Nov 2020

Transcriptome analysis of Curcuma wenyujin from Haikou and Wenzhou and a comparison of the main constituents and related genes of Rhizoma Curcumae

PONE-D-19-24237R2

Dear Dr. Lilan,

We’re pleased to inform you that your manuscript has been judged scientifically suitable for publication and will be formally accepted for publication once it meets all outstanding technical requirements.

Kind regards,

Christian Schönbach, Dr.rer.nat.

Section Editor

PLOS ONE

Additional Editor Comments (optional):

Reviewers' comments:

Reviewer's Responses to Questions

**Comments to the Author**

1. If the authors have adequately addressed your comments raised in a previous round of review and you feel that this manuscript is now acceptable for publication, you may indicate that here to bypass the “Comments to the Author” section, enter your conflict of interest statement in the “Confidential to Editor” section, and submit your "Accept" recommendation.

Reviewer #1: All comments have been addressed

Reviewer #4: All comments have been addressed

2. Is the manuscript technically sound, and do the data support the conclusions?

Reviewer #1: Yes

Reviewer #4: Yes

3. Has the statistical analysis been performed appropriately and rigorously? 

Reviewer #1: Yes

Reviewer #4: N/A

4. Have the authors made all data underlying the findings in their manuscript fully available?

Reviewer #1: Yes

Reviewer #4: Yes

5. Is the manuscript presented in an intelligible fashion and written in standard English?

Reviewer #1: No

Reviewer #4: Yes

6. Review Comments to the Author

Reviewer #1: All the comments have been addressed. Manuscript Transcriptome analysis of Curcuma wenyujin from Haikou and Wenzhou and a comparison of the main constituents and related genes of Rhizoma Curcumae is now acceptable.

Reviewer #4: the revised manuscript has complied all the query of the reviewers and meets all the mandate of the journal.

7. PLOS authors have the option to publish the peer review history of their article (what does this mean?). If published, this will include your full peer review and any attached files.

Reviewer #1: No

Reviewer #4: **Yes: **Enketeswara subudhi

---

## [Editor Report · Acceptance letter]

16 Nov 2020

PONE-D-19-24237R2 

Transcriptome analysis of *Curcuma wenyujin* from Haikou and Wenzhou, and a comparison of the main constituents and related genes of Rhizoma Curcumae 

Dear Dr. Lu:

I'm pleased to inform you that your manuscript has been deemed suitable for publication in PLOS ONE. Congratulations! Your manuscript is now with our production department. 

Kind regards, 

on behalf of

Dr. Christian Schönbach 

Section Editor

PLOS ONE